Spatiotemporal relationships of coyotes and free-ranging domestic cats as indicators of conflict in Culver City, California

Davenport Rebecca N. 1 2 rebeccadavenportn@gmail.com
Weaver Melinda 1
Weiss Katherine C. B. 3
Strauss Eric G. 1
1 Center for Urban Resilience, Loyola Marymount University , Los Angeles, California , United States
2 Department of Forestry and Natural Resources, University of Kentucky , Lexington, Kentucky , United States
3 School of Life Sciences, Arizona State University , Tempe, Arizona , United States
Lambert Max
Electronic publication date: 2022 Oct 7
Publication date: 2022
Volume: 10
Electronic Location ID: e14169
Received 2022 Jan 11; Accepted 2022 Sep 12
Copyright: © 2022 Davenport et al.
Copyright year: 2022
Copyright holder: Davenport et al.
License: This is an open access article distributed under the terms of the Creative Commons Attribution License, which permits unrestricted use, distribution, reproduction and adaptation in any medium and for any purpose provided that it is properly attributed. For attribution, the original author(s), title, publication source (PeerJ) and either DOI or URL of the article must be cited.
License URL: https://creativecommons.org/licenses/by/4.0/

Keywords: Coyote, Domestic cat, Cottontail rabbit, Urban ecology, Southern California, Occupancy, Spatiotemporal, Free-ranging, Camera trap, Conflict

Funding: City of Culver City, California This work is funded by the City of Culver City, California. The funders had no role in study design, data collection and analysis, decision to publish, or preparation of the manuscript.

==============================
As habitat generalists, urban coyote (Canis latrans) populations often utilize an abundance of diverse food sources in cities. Within southern California, domestic cats (Felis catus) comprise a higher proportion of coyote diets than in other studied urban areas throughout the United States. However, it is unclear which ecological factors contribute to higher rates of cat depredation by coyotes in this region. While previous research suggests that coyote presence may have a negative effect on free-ranging domestic cat distributions, few studies have determined whether urban green spaces affect coyote or free-ranging domestic cat occurrence and activity within a predominantly urbanized landscape. We placed 20 remote wildlife cameras across a range of green spaces and residential sites in Culver City, California, an area of Los Angeles County experiencing pronounced coyote-domestic cat conflict. Using data collected across 6 months from 2019–2020, we assessed the influence of green space and prey species (i.e., cottontail rabbits (Sylvilagus spp.) and domestic cats) on coyote habitat use and activity. Coyotes exhibited a preference for sites with higher amounts of green space, while domestic cat habitat use was high throughout our study region. Although cottontail rabbit habitat use was also highly associated with urban green space, neither cottontails nor domestic cats appeared to temporally overlap significantly with coyotes. Unlike other cities where coyotes and domestic cats exhibit strong habitat partitioning across the landscape, domestic cats and coyotes spatially overlapped in green space fragments throughout Culver City. We suggest that this pattern of overlap may be responsible for the frequent cases of domestic cat depredation by coyotes in Culver City.

Introduction

More than four billion people currently live in urban areas, representing over half of the world’s human population (Ritchie & Roser, 2018). As cities continue to expand in size and degree of development (Güneralp et al., 2020), wildlife species affected by urbanization must continuously respond to dramatic changes in their environment. Urbanization often fragments landscapes (Riley et al., 2003; Forman, 2014), which can have subsequent effects on the behavior (Breck et al., 2019; Ellington & Gehrt, 2019), foraging ecology (Fuirst et al., 2018; Smith et al., 2018), and population demography (Graser et al., 2012) of urban wildlife. Large-scale anthropogenic influence can also produce novel community dynamics and trophic interactions by altering the distribution of apex predators and mesocarnivores (Prugh et al., 2009; Newsome & Ripple, 2015; Smith et al., 2018) and introducing novel competitors and prey sources (e.g., domestic cats (Felis catus)) (Kikillus et al., 2017). However, much remains unknown about the extent to which species interact in urban environments, and how these relationships may differ within and among cities.

One mesopredator playing a key role in such novel interactions is the coyote (Canis latrans), whose ranges have expanded with urbanization to occupy most of North and Central America (Hody & Kays, 2018). This is likely due to the ability of coyotes to use human food resources (Fedriani, Fuller & Sauvajot, 2001; Poessel, Gese & Young, 2016; Larson et al., 2020), as well as the extirpation of former apex predators (i.e., wolves (Canis lupus and Canis rufus) and mountain lions (Puma concolor) (Levi & Wilmers, 2012; Newsome & Ripple, 2015)). Coyotes in cities tend to be bolder (Breck et al., 2019), consume novel prey sources (e.g., domestic cats (Felis catus), ornamental fruits, and trash) (Larson et al., 2020), and habituate to people when fed (Young, Hammill & Breck, 2019). Moreover, coyotes fulfill the role of apex predator in many cities (Gehrt & McGraw, 2007), thereby affecting the distribution and abundance of other urban mesocarnivores (Fascione et al., 2004; Greenspan, Nielsen & Cassel, 2018). For example, there is strong evidence that coyotes play a role in reducing free-ranging domestic cat populations (Crooks & Soulé, 1999; Grubbs & Krausman, 2009; Brashares et al., 2010; Cove et al., 2012; Kays et al., 2015). Coyotes are therefore an excellent model organism through which to study the effects of urbanization on predator-prey interactions and relative habitat use.

Due to their popularity as pets, domestic cats are one of the most prevalent introduced species in the world (Dickman, 1996; Medina et al., 2011). In urban systems, free-ranging cats—which we define as domestic cats that are owned and given outdoor access, as well as stray/feral domestic cats (Gehrt et al., 2013; Vanek et al., 2021)—have been responsible for significant levels of predation on native organisms, especially songbirds (Gillies & Clout, 2003; Baker et al., 2008; Dickman, 2009; Santiago-Alarcon & Delgado-V, 2017) and small mammals (Loss, Will & Marra, 2013). Despite the prevalence of free-ranging cats within and around urban landscapes, apex predators and mesocarnivores may exert substantial top-down control on urban domestic cats. For example, free-ranging cats may be a frequent source of prey for coyotes in certain urban contexts (e.g., Grubbs & Krausman, 2009; Larson et al., 2015; Larson et al., 2020). Some wildlife managers may perceive cat depredation as beneficial given the environmental consequences associated with free-ranging cats. However, interactions between cats and coyotes produce strong ethical and social dilemmas (Gramza, Teel & Crooks, 2014; Gramza et al., 2016), especially in cities where cat owners regularly permit their pets to roam outside.

Some studies have quantified the effect of anthropogenic resource availability, including domestic cats, on coyote diet composition in rural and urban areas (e.g., Fedriani, Fuller & Sauvajot, 2001; Poessel, Mock & Breck, 2017). Research suggests that cats account for only a small percentage (0–2%) of coyote diets in various urban systems (e.g., Hernández et al., 2002; Prugh, 2005; Gehrt & McGraw, 2007; Morey, Gese & Gehrt, 2007; Murray et al., 2015; Poessel, Mock & Breck, 2017; Peterson et al., 2021). However, within southern California, domestic cats comprise a significantly higher proportion of coyote diets than elsewhere in the country (Larson et al., 2015; Larson et al., 2020), with up to 20% of surveyed coyote scats in the Los Angeles area containing domestic cat (Larson et al., 2020). It is yet unclear which variables contribute to higher rates of cat depredation by coyotes in the Los Angeles region compared to other areas of the United States.

One factor that may contribute to coyote-cat conflicts in southern California is the relative site use and site overlap between each species. Coyote presence and/or abundance may have a significantly negative effect on free-ranging cat distributions (Crooks & Soulé, 1999; Sims et al., 2008; Cove et al., 2012; Kays et al., 2015), with coyotes preferentially occupying less developed areas across urban to rural gradients, while domestic cats select for more urbanized and residential spaces (Gehrt et al., 2013; Vanek et al., 2021). However, small-scale natural areas within cities, such as urban green spaces, may also influence coyote and free-ranging cat occupancy. Across green spaces in Chicago, cats are more likely to occupy city parks whereas coyotes generally occupy other green spaces, such as golf courses, cemeteries, and natural areas (Gallo et al., 2017). Therefore, patterns of relative habitat use between coyotes and free-ranging cats may differ at points along an urban to rural gradient. Considering elevated rates of coyote-cat conflict in southern California, this work may allow for comparisons of coyote and cat habitat use with other metropolitan areas.

Culver City, California is a highly populated suburb of Los Angeles that has recently experienced frequent cases of domestic cat depredation by coyotes (Culver City Police Department, 2022; S1 File). Given the range of parks, neighborhoods, and developed areas within this narrow urban grid, it is possible that interconnections between urban green space and residential areas facilitate coyote-cat conflicts at a local scale. If this is the case, our findings may better focus management efforts toward areas with a higher probability of conflict. Additionally, this study may address the social implications of domestic cat owners allowing their cats to roam freely outdoors.

In addition to similarities in spatial use, temporal overlap between domestic cats and coyotes could further exacerbate domestic cat mortality in Culver City. Typically, coyotes display nocturnal activity patterns in urban environments, while domestic cats in urban areas are more diurnal (Kays et al., 2015). Given the abnormally high rates of cat depredation by coyotes in southern California (Larson et al., 2015; Larson et al., 2020), we predict that cats in Culver City may exhibit more nocturnal activity, thus demonstrating temporal overlap with coyotes across urban green spaces and residential sites. Additionally, the presence of shared prey species between coyotes and cats, such as cottontail rabbits (Sylvilagus spp.), may contribute to habitat selection by coyotes in ways that inflate their spatial and/or temporal overlap with domestic cats, and therefore opportunities for conflict. Consequently, we also investigated relationships between coyotes and cottontail rabbits throughout our study area.

To better understand coyote and free-ranging cat interactions in Culver City, California, we surveyed coyotes, cats, and cottontail rabbits for 6 months using motion-sensor camera traps. As a noninvasive survey tool, remote wildlife cameras are especially useful when surveying mammalian carnivores, such as coyotes and cats (Ordeñana et al., 2010; Cove et al., 2012; Lombardi et al., 2020). Our study had two main objectives. First, using single-season occupancy models, we sought to determine the relative influence of green space, domestic cat habitat use, and cottontail rabbit habitat use on urban coyote habitat use. Occupancy models are often used to analyze camera trap studies, as they account for imperfect detection and can reveal correlations between species and how they select for habitats (Tobler et al., 2015; Davis et al., 2018; Neilson et al., 2018; Sollmann, 2018). Second, we sought to assess temporal overlap between coyotes and both domestic cats and cottontail rabbits in our study area. We hypothesize that in our study system, coyotes will prefer green spaces, while cats will primarily occupy residential spaces, but that coyote and cat distributions may overlap on temporal scales. We further expect that cottontail rabbit distributions will influence the relative site selection and activity patterns of coyotes, thus mediating coyote-cat conflict. Primarily, this research will assist the Culver City government in establishing effective management protocol for coyotes in this region. However, this case study may also present a feasible approach for other urban areas experiencing human-wildlife conflict. Beyond coyote management, similar studies may further inform how cities regulate the prevalence of free-ranging cats at local urban scales.

Materials and Methods

Study site

Culver City, California, is a city in Los Angeles County with a population of approximately 39,185 people and a total area of 13.31-km2 (Fig. S1). This small region is comprised of several residential areas, a range of local and commercial businesses, and many fragments of green space. There are eight major public parks distributed across Culver City. Culver City Park and Veterans Memorial Park are the largest, covering 41.55 acres and 12.9 acres, respectively. Lindberg Park, Carlson Park, Syd Kronenthal Park, Tellefson Park, Blanco Park, and Hillside Memorial Park range from 1.5 to 4.39 acres (Fig. S2).

Culver City is bordered by Baldwin Hills, a potential point of origin and entry for urban coyotes given its relative green space and proximity to the Kenneth Hahn State Recreation Area (Fig. S2). This city also shares a border with the Inglewood Oil Fields, which covers approximately 1,000 acres of Culver City (History of Inglewood Oil Field, 2017). One of the most distinctive geographic features in the region is Ballona Creek, a 14.16-km watershed that runs from northeast to southwest Culver City and empties into the Santa Monica Bay (Fig. S2). Access to Ballona Creek is mostly limited to a bike path that begins at Syd Kronenthal Park in east Culver City and runs past Culver City Park and Lindberg Park. This geographic feature may serve as a necessary water source and means of connectivity for urban wildlife.

Camera trap surveys

From December 2019 through June 2020, we conducted a camera trap survey in Culver City, California using 20 non-baited wildlife cameras (BTC 6HDX Dark Ops 940; Browning Trail Cameras, Birmingham, AL, USA) for a total of 3,736 sampling nights across all cameras (Fig. 1). Twenty remote wildlife cameras have been identified as the minimum sample size needed to assess the occupancy of common species (Kays et al., 2020). Cottontail rabbits occur widely across the urban-rural gradient in southern California (Larson et al., 2020) and are frequently cited as a common prey species for coyotes (e.g., MacCracken & Hansen, 1987; Swingen, DePerno & Moorman, 2015; Poessel, Mock & Breck, 2017). Similarly, the number of coyote sightings reported by community members and cat mortality events within Culver City neighborhoods, as well as the high proportion of cat remains in coyote diet within southern California (Larson et al., 2020), suggests that the species assessed in our study are common enough to support our sample size.

Figure 1 Map of 20 remote wildlife cameras across Culver City, California.

Inset map illustrates the location of camera sites within the broader context of Los Angeles County. Sources: Esri, Maxar, GeoEye, Earthstar Geographics, CNES/Airbus DS, USDA, USGS, AeroGRID, IGN, and the GIS User Community.

When possible, cameras were positioned at roughly knee-height, ~51 cm high. However, several cameras were placed at heights above or below this average due to limitations in the urban landscape (e.g., lack of stable attachment site or sloped terrain at knee-height). To test for the potential influence of camera height on the probability of coyote, free-ranging cat, and cottontail rabbit detection, camera height was included as a covariate on detection probability in our analyses (see Statistical Analyses).

Given the fine scale of this study, cameras were positioned >50-m apart (Fig. 1). Though 26 cameras were originally placed throughout our study region, if more than one camera was present within the same 50-m buffer zone, we randomly selected one camera to be used for the analysis (Table S1). While it is standard for camera trap studies to position cameras at least one home range diameter apart to avoid detection of the same individuals (Sollmann, 2018), our work was conducted under the assumption that Culver City lies within the home range of 1–2 packs of coyotes. Coyote trapping, ear tagging, and radio collaring throughout the study area (M Weaver, 2020, personal communication) has confirmed that individuals detected across multiple sites belong to the same pack. Thus, our study aimed to assess local habitat selection of a small number of coyotes rather than the selection of multiple populations of coyotes. A 50-m buffer zone was therefore used to assess variation in wildlife site use at the finer scale of neighboring green spaces and residential areas.

Community science reports (S1 File) between July 2013 and December 2019 indicated that coyote sightings and cat deaths were predominantly located within two estimated quadrants of Culver City (Fig. 2). Given that the initial aim of this study was to assist the Culver City local government in mitigating instances of human-wildlife conflict, we intentionally selected sites within these broad zones. We recognize that camera placement in areas with known coyote presence positively biased our sites toward both coyote and cat detections, given that reports of coyote sightings often corresponded to evidence of cat mortality (S1 File). However, coyotes and cats were not known to occur specifically at each site. We distributed cameras across a range of green spaces and residential sites within these general hotspots (Fig. 1). For example, at least one camera was deployed at each of the major green spaces within these zones. This included urban parks and other potential wildlife corridors, such as the Ballona Creek bike path and storm drains. Public parks were included to assess the expected correlation between green space and coyote site use based on evidence from the literature (Gehrt et al., 2013).

Figure 2 Heat map of coyote sightings in Culver City between July 2013 and December 2019.

Red circles indicate locations of coyote sightings across Culver City. Yellow and red areas on the map highlight hotspots of coyote sightings due to clustered data points. Blue areas surround regions with fewer coyote sightings. Sources: Esri, HERE, Garmin, Intermap, increment P Corp., GEBCO, USGS, FAO, NPS, NRCAN, GeoBase, IGN, Kadaster NL, Ordnance Survey, Esri.

Several cameras were also distributed throughout residential neighborhoods. In some cases, concerned residents directly contacted us and granted us access to their backyards for coyote monitoring. As a result, a few of our residential sites fell outside of the coyote hotspot boundaries. In an effort to capture as many green spaces as possible, we also selected non-hotspot sites within Lindberg Park, Veterans Park, Culver City Park, and adjacent to Baldwin Hills. Nevertheless, it should be noted that site selection was not randomized and that results must be interpreted accordingly. Altogether, cameras were evenly distributed between urban green spaces and highly urbanized/residential sites (Fig. S3). All camera locations were approved by the City of Culver City, California (permit #32000041).

We used the Sanderson’s CameraSweet to sort and analyze photo data (Sanderson & Harris, 2013). Independent observers sorted each photograph, followed by an expert researcher who performed quality checks for each data set to further validate species identifications prior to analysis. All observers were trained and checked for accuracy prior to sorting photos. From December 2019 through June 2020, we collected 892 free-ranging cat images, 510 coyote images, and 1,747 cottontail rabbit images. We considered photographs of each species taken at individual sites to be independent if the images were captured more than 30 min apart (Linkie & Ridout, 2011; Havmøller et al., 2020).

Geographic information system analyses

To calculate the degree of green space versus residential space across sites, we used ArcMap (Version 10.6.1; ESRI, Redlands, CA, USA) to first construct a 150-m radius around each site. This buffer zone is used for distinguishing landscape types at a smaller scale, thereby classifying the vegetation directly surrounding each camera trap (Ordeñana et al., 2010). We recognize that broadening our original 50-m radius between cameras to 150-m buffer zones for this landscape assessment resulted in substantial overlap between some of the neighboring sites. However, it was important to capture a broader radius of landscape features that may have contributed to the detection of coyotes and cats in our study area. Moreover, recent work suggests that overlapping landscapes may neither violate independence between sites nor cause pseudoreplication (Zuckerberg et al., 2012; Zuckerberg et al., 2020). Nevertheless, we ensured that sites with overlapping buffers did not have correlated green space values before proceeding with our analyses (R2 = 0.0082) (Fig. S3).

Using the tessellation function in ArcMap (Version 10.6.1; ESRI, Redlands, CA, USA), we constructed a continuous non-overlapping hexagon layer for each of the 20 sites (McDonald et al., 2008). Each hexagon was 100-m2 with an average of 334 hexagons per site. Using satellite view, we then counted the number of hexagons reflective of green space versus residential space. Green space was classified as vegetation, public parks, natural areas, baseball fields, cemeteries, oil fields, dirt patches, and water sources (including Ballona Creek). Residential space included neighborhoods, buildings, roadways, and other man-made features. If a hexagon reflected some quantity of green space and residential space, we classified it by majority. Given that each hexagon was categorized into one of these two groups, our covariate of green space was considered the inverse of residential/urbanized space. Values of green space were calculated as the proportion of green space hexagons (mean = 0.563, sd = 0.204) and then standardized to have zero mean and unit variance across the 20 sites (Table S1).

Single-species occupancy analyses

To assess the influence of relative green space, domestic cat distributions, and cottontail rabbit distributions on the probability of site use by coyotes, as well as to identify the distribution of domestic cats and cottontails more broadly, we developed a series of single-species occupancy models (MacKenzie et al., 2018). Since sampling locations may have lacked independence due to their close proximity, we defined and interpreted occupancy as the relative habitat use of each species (Magle et al., 2021). We tested our habitat covariate of green space on occupancy (Ψ) and evaluated the influence of camera height on detection probability (p). Although we intended to run occupancy models for free-ranging cats, cats were detected in 17 out of 20 sites, leaving minimal variation to be explained by landscape patterns and coyote occupancy. Instead of performing the full suite of occupancy models on free-ranging cats, we ran only the null intercept model with no covariates to obtain a value for cat occupancy and detection probability. Although cat occupancy was high throughout our study area, we thought it worthwhile to examine variation in the number of cats detected at each site per 30-min period (NumCat). Therefore, for the coyote and rabbit occupancy analyses, the number of cat photos taken at a site within a 30-min period (NumCat) was included as an additional covariate on Ψ and standardized to have zero mean and unit variance (Table S1). Considering that free-ranging cats were not individually identified across the study period, this metric was designed to indicate relative site use of a certain number of individuals as opposed to cat abundance. Hence, greater NumCat values at a given site served as a proxy for greater relative site use compared to other camera trap locations. 30-min cutoffs are standard for identifying independence between detections (Linkie & Ridout, 2011; Havmøller et al., 2020), especially for larger data sets when individual animals cannot be identified.

We considered including an equivalent covariate for the number of rabbits detected at each site (NumRab) for the coyote occupancy models. However, upon running a Pearson’s correlation matrix to ensure that our variables were not too highly correlated, we found NumRab and green space to have a correlation of 0.623 (Table S2). Given the frequently accepted cutoff of +/− 0.5 (e.g., Kays & Parsons, 2014; Colborn et al., 2020), we decided to remove NumRab as a covariate on Ψ. This high correlation suggests that the models would be unable to differentiate the respective influences of environmental factors (e.g., green space/residential space) versus prey availability (e.g., rabbit detections) on coyote occupancy. However, contrary to free-ranging cats, rabbits were only detected at seven of the sites throughout the study area. Therefore, we deemed it acceptable to run the full set of occupancy models on rabbits to test for a correlation between rabbit occupancy and green space.

For both the coyote and rabbit occupancy models, no other variables had a correlation equal to or above +/− 0.5. We modeled all eight possible combinations of Ψ and p in our single-species occupancy models for coyotes and cottontail rabbits, which included a null intercept model (Table 1). Candidate models were then ranked using values of the Akaike Information Criterion corrected for small sample size (AICc) (Burnham & Anderson, 2002). We also calculated relative variable importance (w+), which is the sum of AICc weights for all models containing a given variable (Burnham & Anderson, 2002). Occupancy modeling was performed using R version 4.0.3 using package ‘RMark’ (Laake, 2013; R Core Team, 2020).

Table 1 Eight coyote occupancy models (including null models) with all possible combinations of covariates.

Model	npar	AICc	Δ AICc	Weight	Deviance	
Ψ(Greenspace), p(CamHeight)	4	116.789	0	0.413	106.123	
Ψ(Greenspace), p(.)	3	117.062	0.273	0.360	109.562	
Ψ(NumCat + Greenspace), p(.)	4	119.527	2.738	0.105	108.861	
Ψ(NumCat + Greenspace), p(CamHeight)	5	119.717	2.928	0.096	105.43	
Ψ(.), p(CamHeight)	3	123.702	6.912	0.013	116.202	
Ψ(.), p(.)	2	124.522	7.733	0.009	42.310	
Ψ(NumCat), p(CamHeight)	4	126.852	10.062	0.003	116.185	
Ψ(NumCat), p(.)	3	127.291	10.501	0.002	119.791	
Note:

Green space and the number of cat photos per site (NumCat) were tested on occupancy (Ψ), while camera height (CamHeight) was tested on coyote detection (p).

Activity pattern analyses

Using data from the Sanderson CameraSweet output file, each independent photo per 30-min interval was assigned the median time of the hour in which the animal was active (i.e., an individual detected between 04:00–05:00 was estimated to be active at 04:30). Although the Sanderson’s CameraSweet analysis did not produce exact activity values, we considered these estimates to be sufficient considering each interval represented only two 30-min periods. Using the package ‘overlap’ in R (Linkie & Ridout, 2011; Kamler et al., 2020), we constructed kernel density estimation curves to depict the activity patterns of coyotes, cats, and rabbits. We then quantified the temporal overlap of coyotes and free-ranging cats, as well as coyotes and cottontail rabbits. We estimated the coefficient of overlap (Δ), or the shaded area under the kernel density curves (Soultan, Attum & Lahue, 2021), using the nonparametric estimator Δ4, as each species assessed had a sample size exceeding 50 (Havmøller et al., 2020; Kamler et al., 2020). To calculate bootstrap percentile confidence intervals for Δ4, we used 10,000 bootstrap samples (Havmøller et al., 2020). Activity patterns were considered significantly different from one another if the upper bound of the 95% confidence interval was <0.90 (Lewis et al., 2021).

Results

Single-species occupancy models

For coyotes, estimated occupancy (Ψ) was 0.516 (se = 0.152, 95% CI [0.244–0.779]), while estimated detection probability (p) was 0.472 (se = 0.064, 95% CI [0.350–0.598]). Green space was included as a predictor of coyote occupancy in both of the highest ranked models, with weights of 0.413 and 0.360, respectively (Table 1). Therefore, we used the top model to draw inferences from the species data. The β estimate for green space (β = 2.163, se = 0.973, 95% CI [0.255–4.070]) revealed a positive and informative relationship with coyote occupancy (i.e., 95% Confidence Intervals (CIs) did not overlap zero). Similarly, green space had a relative variable importance (w+) of 0.973, indicating that of the two covariates assessed on Ψ, green space was the most important covariate included in the coyote occupancy models (Table S3). On the other hand, the number of cat detections per site (NumCat) had a relatively low w+ of 0.205 (Table S3) and was not included as a predictor of coyote occupancy in the top models (Table 1). Consequently, the relationship between NumCat and coyote occupancy was considered nominal and uninformative.

Camera height had a w+ of 0.524 (Table S3) and a β estimate of 0.481, suggesting a positive relationship with coyote probability of detection. However, the lower and upper bounds of the 95% CIs [−0.038 to 0.999] overlapped zero, indicating that this covariate was not informative for probability of detection, as the direction of the relationship could not be determined with high confidence.

For cottontail rabbits, Ψ was 0.259 (se = 0.132, 95% CI [0.084–0.573]), while p was 0.570 (se = 0.071, 95% CI [0.428–0.701]). Similar to coyotes, the highest ranked model also included green space as a predictor of rabbit occupancy (Table 2). The β estimate of green space (β = 2.043, se = 0.880, 95% CI [0.318–3.767]) had a strong positive relationship with rabbit occupancy and was modeled as the most influential covariate on occupancy, with a w+ of 0.958 (Table S4). The 95% CIs did not overlap zero, suggesting that this relationship was informative. On the other hand, NumCat was not included as a predictor of Ψ in the top models (Table 2) and had a relatively low w+ of 0.191 (Table S4), suggesting that this covariate was uninformative for rabbit occupancy. Camera height was not included in the top models (Table 2) and had a relatively low w+ of 0.208 (Table S4), suggesting that this covariate was not informative for rabbit probability of detection.

Table 2 Eight rabbit occupancy models (including null models) with all possible combinations of covariates.

Green space and the number of cat photos per site (NumCat) were tested on occupancy (Ψ), while camera height (CamHeight) was tested on rabbit detection (p).

Model	Npar	AICc	DeltaAICc	Weight	Deviance	
Ψ(Greenspace), p(.)	3	90.847	0	0.619	83.347	
Ψ(Greenspace), p(CamHeight)	4	93.547	2.700	0.160	82.880	
Ψ(NumCat + Greenspace), p(.)	4	93.803	2.956	0.141	83.136	
Ψ(NumCat + Greenspace), p(CamHeight)	5	96.433	5.587	0.038	82.148	
Ψ(.), p(.)	2	97.423	6.576	0.023	33.959	
Ψ(NumCat), p(.)	3	99.213	8.367	0.009	91.713	
Ψ(.), p(CamHeight)	3	99.761	8.914	0.007	92.261	
Ψ(NumCat), p(CamHeight)	4	101.904	11.057	0.002	91.237	

The null intercept model for free-ranging cats estimated Ψ as 0.882 (se = 0.082, 95% CI [0.614–0.972]) and p as 0.445 (se = 0.048, 95% CI [0.353–0.540]).

Activity analyses

Both cats and rabbits showed statistically different activity patterns from coyotes. In both cases, the upper bound of the 95% CIs for the coefficient of overlap was <0.90 (e.g., Lewis et al., 2021) (Table 3). Coyotes in Culver City displayed a primarily nocturnal activity pattern with peaks around 04:00 and 24:00 (Fig. 3). Cats were most active between 00:00–06:00 and 18:00–24:00, with the most prominent peak at around 20:00 (Fig. 3A). Cats displayed slight diurnal activity, while coyote activity was largely absent during diurnal time periods. Rabbits were the most active from 22:00–7:00, suggesting both nocturnal and crepuscular activity patterns and marginal diurnal activity (Fig. 3B).

Table 3 Coefficient of overlap between activity patterns of coyotes (N = 148) and prey species (free-ranging cats and cottontail rabbits) in Culver City, California, USA.

Species	N	Δ	Lower CI	Upper CI	
Cat	301	0.710	0.602	0.771	
Rabbit	433	0.722	0.617	0.759	
Note:

N, number of independent photos of species; Δ, overlap estimate; CI, 95% confidence interval.

Figure 3 Overlap in daily activity patterns between (A) coyotes and cats, and (B) coyotes and rabbits.

Coyote patterns are displayed with a solid line, while prey species are displayed with a dashed line. Time of day is presented on the x-axis, and kernel density activity is displayed on the y-axis. The shaded area under the kernel density curves represents the coefficient of overlap (Δ) between coyote and prey activity patterns.

Discussion

This study aimed to understand spatiotemporal interactions between urban coyotes and free-ranging cats in Culver City, California. We examined potential factors influencing coyote relative habitat use, including urban green space, free-ranging cat habitat use, and cottontail rabbit habitat use. Our single-season occupancy models for coyotes and cottontails tested covariates of green space and number of cat detections per site (NumCat) on species occupancy, while camera height was tested on probability of species detection. Coyote and rabbit occupancy were best predicted by increasing levels of green space at sites compared to the availability of cats or no covariates at all (null model). This aligned with our original hypothesis that coyote site use would be positively associated with green space. However, based on previous findings, we predicted that cats would primarily occupy residential spaces. Instead, we found that free-ranging cats were present in both residential spaces and green spaces, thus resulting in potential spatial overlap between these species. We also assessed temporal overlap between coyotes and two likely prey species – domestic cats and cottontail rabbits. Activity patterns significantly differed between coyotes and both prey species, although the analyses indicated some degree of overlap. This partially deviated from our hypothesis that temporal overlap may contribute to elevated rates of coyote-cat conflict in Culver City.

Coyote site use patterns aligned with those of other studies, which found coyotes to show a preference for natural areas within a matrix of urbanization (e.g., Gehrt et al., 2013; Lombardi et al., 2017; Vanek et al., 2021). Given that coyotes prefer green spaces as human development simultaneously expands, it is possible that coyotes select for habitat fragments with more green space as a means of navigating through residential neighborhoods and urbanized regions. Green space is also an essential factor for coyote dens and access to water (Way et al., 2002; Schmidly & Bradley, 2016). This might explain coyotes in our study using green spaces, such as Ballona Creek or in regions surrounding Baldwin Hills. However, it may be useful for future studies to classify green spaces into specific types based on landscape characteristics, as coyote site use may be specifically associated with certain green space attributes.

It is important to note that site selection may have positively biased coyote detections within Culver City green spaces, as cameras were generally placed throughout large hotspot zones of coyote sightings (see Camera Trap Surveys; Fig. 2). However, this positive bias would have likely affected detections in both green spaces and residential areas, as green space values varied greatly between sites (Fig. S3). In addition, a few cameras were positioned in green spaces and backyards outside of the hotspot zones (Fig. 1), which may have mitigated some possible bias. Therefore, while overall estimates of coyote occupancy (Ψ = 0.516) and probability of detection (p = 0.472) may be slightly inflated, we still conclude that coyote relative habitat use is positively associated with green space.

Although coyotes in Culver City appear to primarily use green spaces, it is unclear whether their site use corresponds more heavily with landscape attributes (e.g., water and den access) or instead with prey availability. If food items are scarce or unavailable in natural areas, coyotes may have to occasionally travel through neighborhoods to obtain resources (Grinder & Krausman, 2001; Gehrt, Brown & Anchor, 2011). This may be the case in Culver City, where there is a lack of full continuity between forested habitats. However, other studies have found that coyotes will primarily forage within green spaces, if such areas are widely available (Gehrt & Riley, 2010). To investigate this possibility, our occupancy models and activity analyses examined if domestic cats and cottontail rabbits are available prey sources for coyotes in urban green spaces. Several studies have classified cottontail rabbits as a common prey species for coyotes (e.g., MacCracken & Hansen, 1987; Swingen, DePerno & Moorman, 2015; Poessel, Mock & Breck, 2017). Our analyses revealed that rabbit site use is also positively correlated with green space. Therefore, high coyote site use in urban green spaces may correspond with higher cottontail rabbit availability in these areas compared with residential zones.

However, prey availability requires both spatial and temporal overlap. While selection for green space by both species may indicate some degree of spatial overlap, our activity analyses sought to examine if coyotes and rabbits overlap on temporal scales. Although both species appeared to exhibit similar nocturnal/crepuscular patterns, their overall activity patterns were statistically different (Lewis et al., 2021). It is possible that rabbits in Culver City have adjusted their activity to avoid coyotes, especially considering that both species show a strong preference for particular sites across a narrow local gradient. Nevertheless, statistical significance does not necessarily equate to biological significance. While the peaks of activity may vary, there still appears to be substantial temporal overlap between these species (Fig. 3B). Based on these spatiotemporal relationships, it is possible that cottontail rabbits are viable prey sources for coyotes in Culver City.

Free-ranging domestic cat activity patterns were also classified as significantly different from coyotes, yet there is already substantial evidence of cat depredation by coyotes in southern California (Larson et al., 2015; Larson et al., 2020). Photographs from our camera trap surveys provide further evidence of cat depredation by coyotes (e.g., Fig. 4). Domestic cats and coyotes likely experience increased temporal overlap in Culver City compared to other cities, as cats in Culver City were more nocturnal than in other urban studies (e.g., Kays et al., 2015; Vanek et al., 2021). Therefore, while cats may be exhibiting patterns of avoidance, they are likely viable prey sources for coyotes, especially compared to other cities. However, the nonrandomized and positively biased site selection in this study (see Camera Trap Surveys) may limit inferences regarding temporal overlap. Considering that several cameras were positioned within general areas where coyote sightings and cat deaths had been reported (Fig. 2), it is possible that this sample does not reflect true activity patterns across the city. Perhaps cats are more active and/or nocturnal in these neighborhoods and parks compared to other areas. However, this does seem improbable considering the small size of Culver City (13.31-km2; Fig. S1) and that this region likely encompasses a single population of coyotes and cats. Nevertheless, future studies are needed to comprehensively assess the activity patterns of both species across a random sample of green spaces and residential sites. Such research could also assess coyote diet composition to determine if coyote diet in Culver City aligns with the elevated proportion of domestic cat remains reported in larger-scale southern California studies (Larson et al., 2015; Larson et al., 2020).

Figure 4 Camera trap photograph of a coyote carrying a domestic cat in its mouth.

Since cat occupancy was high across our study system (Ψ = 0.882), we wanted to evaluate whether the relative site use of free-ranging cat detections affected coyote and/or rabbit occupancy (covariate of NumCat). It is important to recognize that neither high occupancy nor greater NumCat values serve as a proxy for free-ranging cat abundances. High occupancy of cats across Culver City could have been due to repeated sampling of the same individuals between sites in close proximity. However, free-ranging cats in urban areas typically have smaller home ranges than cats in rural areas (Horn et al., 2011; Hall et al., 2016). Additionally, owned cats often have home ranges as small as 1–3 hectares (Horn et al., 2011; Castañeda et al., 2019; Pirie, Thomas & Fellowes, 2022), although unowned cat home ranges can extend up to 157 hectares (Horn et al., 2011). If most of the free-ranging cats in Culver City are owned, then 50-m radius buffers (area = 0.785 ha) between cameras may be sufficient in establishing some degree of independence between sites for cat detections. Further studies are required to determine the ownership status of Culver City free-ranging cats and estimate home range size. Given that we were unable to identify individual cats at our sites, future work may confirm or deny that the high occupancy of cats in Culver City corresponds to an abundance of cats across the landscape. Still, we can conclude that the cats detected in our study are widely present across the selected sites and appear to use a combination of green space and residential habitat.

Interestingly, neither coyote nor rabbit relative habitat use were associated with the relative site use of free-ranging cats (NumCat). One possible explanation is that Culver City coyotes do not prey on domestic cats at a degree that might influence coyote site use patterns. However, there is evidence to suggest that coyotes in this region heavily prey upon free-ranging cats. Community science reports to the Culver City government include several direct observations of cat depredation by coyotes (S1 File), as well as indirect evidence of mortality through cat necropsies and nearby coyote sightings. Similarly, the high proportion of cat remains in the diet of southern California coyotes (Larson et al., 2015; Larson et al., 2020) may suggest that elevated rates of cat depredation by coyotes are a general pattern throughout this region of the state.

Based on this evidence, it may be the case that coyotes prey quite heavily on free-ranging cats without preferentially occupying sites with higher cat site use. Perhaps available habitat (e.g., green space) influences spatial distributions of Culver City coyotes more than prey availability. Coyotes have previously been cited as optimal foragers (MacCracken & Hansen, 1987; Hernández et al., 2002). If they are able to locate food regardless of availability of prey, green space may be their main limiting factor. However, it remains unclear whether coyotes select for green space in Culver City independent of its positive association with cottontail rabbit site use. Lagomorphs are known to be a primary prey source for coyotes (Clark, 1972; Andelt et al., 1987; Hernández, Delibes & Hiraldo, 1994). In Los Angeles in particular, cottontail rabbits have been found to account for 18.04% of coyote scats (Larson et al., 2020). Moreover, rabbits may fail to spatially avoid coyotes or to display heightened vigilance when coyotes are present (Gallo et al., 2017). Such behavioral tendencies may contribute to cottontail rabbits being a common prey item for coyotes, even in urban areas.

Similarly, domestic cats in our study were detected in all but one of the sites in which coyotes were detected. Even though our study did not assess the relative abundances of free-ranging cats across Culver City, we know that cats were at least present in sites with moderate to high levels of green space cover. Our camera traps also captured evidence of cottontail rabbit depredation by a free-ranging cat (Fig. 5). Therefore, it remains possible that the presence of shared prey species between coyotes and cats influence habitat selection by both mesopredators. Based on these observations, patterns of site use overlap in Culver City appear to dramatically differ from other cityscapes, where coyotes restrict cats to developed areas through intraguild competition (Crooks & Soulé, 1999; Sims et al., 2008; Cove et al., 2012; Kays et al., 2015). In Chicago, for example, cats and coyotes partition the landscape, with minimal overlap in home range (Gehrt et al., 2013). Cats were presumed to avoid coyotes by remaining on the periphery of natural habitat fragments (Gehrt et al., 2013). In Culver City, the wide spatial distribution of free-ranging cats may partially explain why cats comprise a disproportionate percentage of coyote diets compared to other urban landscapes across the country. However, Culver City serves as only a case study for the greater Los Angeles area. Further diet studies are necessary at a local scale to confirm the predicted availability of free-ranging cats as a prey species. If locating prey is not particularly difficult for Culver City coyotes, then these individuals may be able to select for green space fragments while still encountering free-ranging cats.

Figure 5 Camera trap photograph of a domestic cat holding a cottontail rabbit in its mouth.

We must again acknowledge that positive bias in the study design may have skewed site use patterns and detection probabilities of free-ranging cats. It is possible that cats at these sites are not representative of the relative site use of the entire free-ranging cat population. Perhaps a random sample of cats may have revealed greater habitat partitioning with urban coyotes. While these speculations cannot be entirely dismissed without further studies, we do note that cats were detected in backyards and green spaces apart from the hotspot zones of coyote sightings and cat deaths. For example, cats used sites in Culver City Park, Veterans Park, and habitat adjacent to Baldwin Hills. Given that these green space sites were more randomly distributed across Culver City than clustered hotspot sites, these observations suggest that spatial overlap between free-ranging cats and coyotes in green space sites was not merely a product of study design.

One of the primary goals of this research was to assist the Culver City government in establishing effective management protocol for coyotes in this region. Based on incidental sightings of cat depredation by coyotes, residents have a common perception that coyotes repeatedly seek access to cats or alternative food sources near Culver City residences. On the contrary, this study suggests that coyote relative habitat use is concentrated within urban green spaces. Based on these findings, management efforts may better monitor coyote behavior and ecology by radio collaring and tracking individuals in these areas.

Although much attention is directed toward control of “problem” coyotes, these results may also inform city management of free-ranging cats. While coyote relative site use was positively associated with green space, domestic cats were detected in both residential areas and green space sites. Spatial overlap combined with increased nocturnality of free-ranging cats suggest that coyote-cat relationships in Culver City substantially differ from other urban areas (e.g., Gehrt et al., 2013; Vanek et al., 2021). Given that free-ranging cats are generally highly invasive predators that threaten a range of native wildlife (Gillies & Clout, 2003; Baker et al., 2008; Dickman, 2009; Loss, Will & Marra, 2013; Santiago-Alarcon & Delgado-V, 2017), we recommend that future management efforts consider possible restrictions or control measures for cats in this area. If coyotes have alternative prey sources available in their preferred habitat, such as cottontail rabbits, then restrictions on outdoor cats could potentially result in fewer instances of cat mortality. Social science research on public risk perceptions and attitudes toward free-ranging cats (Gramza et al., 2016) should be used to navigate the complex social dimensions of various control measures. Our results indicate that spatiotemporal relationships of coyotes and free-ranging cats may vary at local scales. Thus, we hope that this case study provides a strong framework and methodological approach for other cities experiencing human-wildlife conflict.

Conclusions

As urbanization continues to encroach on green space in southern California, the resulting influx of coyotes to developed areas may facilitate human-wildlife conflict. In this case, coyotes have been linked to frequent cases of cat depredation in Culver City, California. Using a local camera trap analysis within a predominantly urban landscape, we propose that spatiotemporal patterns of coyotes and cats in Culver City may distinguish conflict in this area from other urban landscapes. Our occupancy models revealed a positive correlation between coyote habitat use and green space, while cats were instead widely detected across both developed areas and natural habitat fragments. This lack of landscape partitioning may, in combination with additional demographic factors and geographical features, be responsible for the high percentage of cat depredation events reported in Culver City. Our study will serve to focus future research toward important differences in the site use of free-ranging cats in Culver City compared to other cities. Future management may redirect some attention toward the social implications of permitting domestic cats to freely roam in Los Angeles.

Supplemental Information

Supplemental Information 1 Culver City positioned within the broader Los Angeles area of southern California.

Sources: Esri, Airbus DS, USGS, NGA, NASA, CGIAR, N Robinson, NCEAS, NLS, OS, NMA, Geodatastyrelsen, Rijkswaterstaat, GSA, Geoland, FEMA.

Click here for additional data file.

Supplemental Information 2 Relevant geographic features and landmarks in Culver City, California.

(A) Ballona Creek. (B) Blanco Park. (C) Carlson Park. (D) Culver City Park. (E) Holy Cross Catholic Cemetery. (F) Inglewood Oil Fields. (G) Kenneth Hahn State Recreation Area. (H) Lindberg Park. (I) Syd Kronenthal Park. (J) Tellefson Park. (K) Veterans Memorial Park. Sources: Esri, Airbus DS, USGS, NGA, NASA, CGIAR, N Robinson, NCEAS, NLS, OS, NMA, Geodatastyrelsen, Rijkswaterstaat, GSA, Geoland, FEMA.

Click here for additional data file.

Supplemental Information 3 Distribution of standardized values of green space across all 20 sites (R2 = 0.0082).

Nine sites have positive values of green space while eleven sites have negative values of green space.

Click here for additional data file.

Supplemental Information 4 Covariates associated with each of the 20 camera trap locations.

Camera height, proportion of green space, and frequency of cat detections were standardized using z-scores.

Click here for additional data file.

Supplemental Information 5 Pearson’s correlation matrix of covariates for occupancy models.

NumRab, number of rabbits detected at each site. NumCat, number of cats detected at each site.

Click here for additional data file.

Supplemental Information 6 Variable importance value (VIV) calculations for each covariate, ranked in order of their relative weights for the coyote occupancy models.

NumCat, number of cats detected at each site.

Click here for additional data file.

Supplemental Information 7 Variable importance value (VIV) calculations for each covariate, ranked in order of their relative weights for the rabbit occupancy models.

NumCat, number of cats detected at each site.

Click here for additional data file.

Supplemental Information 8 Citizen science reports of coyote sightings and cat deaths.

Excel spreadsheet of citizen science reports of coyote sightings and cat deaths reported to Culver City.

Click here for additional data file.

Supplemental Information 9 Cat, rabbit, and coyote activity from camera trap data.

Each line represents an individual photograph, and indicates at what time the animal was active. The dataset also classifies each individual photograph as within a green space study site or not.

Click here for additional data file.

Supplemental Information 10 R script for activity analyses of cats, rabbits, and coyotes.

This R script includes annotated methods for determining overlap in activity patterns between cats and coyotes versus cats and rabbits. It provides the relevant code for creating figures to illustrate the overlap in activity.

Click here for additional data file.

Supplemental Information 11 Model-averaged beta estimates, occupancy, and probability of detection.

This spreadsheet shows output values from the occupancy models for coyotes and rabbits, as well as calculations for model-averaged beta estimates, occupancy, and probability of detection.

Click here for additional data file.

Supplemental Information 12 Covariates for occupancy models of coyotes and rabbits.

This dataset includes information about each site and their associated values of green space, camera height, number of cats, and number of rabbits.

Click here for additional data file.

Supplemental Information 13 Occupancy matrix input file for coyote occupancy model.

This text file includes an occupancy matrix, with one value for each 30 day period of the study per each of the 20 sites. The occupancy matrix is followed by corresponding covariates.

Click here for additional data file.

Supplemental Information 14 Rabbit occupancy model R script.

This R script provides annotated code for the rabbit single-species occupancy models.

Click here for additional data file.

Supplemental Information 15 Coyote occupancy model R script.

This R script provides annotated code for the coyote single-species occupancy models.

Click here for additional data file.

Supplemental Information 16 Occupancy matrix file for rabbit occupancy model.

This text file includes an occupancy matrix, with one value for each 30 day period of the study per each of the 20 sites. The occupancy matrix is followed by corresponding covariates.

Click here for additional data file.

We are grateful to the City of Culver City and the Center for Urban Resilience (CURes) for providing support, resources, and knowledge to our research team throughout the project. We are also grateful to our dedicated and diligent team of undergraduate students in the Weaver Lab at Loyola Marymount University (LMU), each of whom greatly assisted with data processing. Specifically, we would like to acknowledge Katherine Arakkal, Anna Maria Brodkey, Julia Burke, Marceline Burnett, Belen Carrasco-Cazares, Gwyneth Garramone, Matthew Goddard, Madina Inagambaeva, Abby Marich, Anna Monterastelli, Sarah O’Riordan, Advait Prasad, Jaime Luis Villa, and Ian Wright.

Additional Information and Declarations

Competing Interests

Author Contributions

Animal Ethics

Field Study Permissions

Data Availability

The authors declare that they have no competing interests.

Rebecca N. Davenport analyzed the data, prepared figures and/or tables, authored or reviewed drafts of the article, and approved the final draft.

Melinda Weaver conceived and designed the experiments, performed the experiments, authored or reviewed drafts of the article, and approved the final draft.

Katherine C. B. Weiss analyzed the data, authored or reviewed drafts of the article, and approved the final draft.

Eric G. Strauss conceived and designed the experiments, authored or reviewed drafts of the article, and approved the final draft.

The following information was supplied relating to ethical approvals (i.e., approving body and any reference numbers):

Our work did not require IACUC approval, as no animals were handled for our research. Camera traps were deployed and data was collected remotely.

The following information was supplied relating to field study approvals (i.e., approving body and any reference numbers):

The City of Culver City approved the deployment of camera traps across Culver City, California (32000041).

The following information was supplied regarding data availability:

The raw data and R script are available in the Supplemental Files.

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
