# Peer review of "Spatiotemporal relationships of coyotes and free-ranging domestic cats as indicators of conflict in Culver City, California"

_PeerJ, doi:10.7717/peerj.14169_

## Round 0.1 · original submission · Major Revisions

Your manuscript has been assessed by three qualified reviewers. All reviewers are enthusiastic about your study. Even so, the reviewers highlighted a number of issues that need to be addressed. In particular, all reviewers had important questions about the study design and analysis and how these issues impact your inferences. Some of these issues may be cleared up with added clarity but other issues may require re-analysis and rewriting portions of the manuscript. The reviewers' other issues largely surround clarity of the writing and some tinkering in the introduction and discussion

Please let me know if you have any questions and I look forward to receiving a revised version of your manuscript.

Reviewer 1 ·

Basic reporting

See uploaded PDF, but some of the methodology is a bit unclear. Literature presented is sufficient and writing is clear and relatively concise throughout. Raw data/R code is well explained, that being said if the author wishes to provide all raw data necessary for re-running the analyses, I don't believe everything was included.

See uploaded PDF for commentary on figures and minor changes and figure/table captions could be a bit better elaborated. (feedback also copied below)

Line 20: in other studied urban areas

Introduction
The introduction is well-written. It does have most pertinent information, but it feels like it is lacking in a reason for this study. If you were to reframe your introduction more on the lines of why we care about this study, because it is important, and allude to some of the more import potential implications while discussing the hypotheses/background, I think it would be much more compelling.

Line 103: What makes you predict increased nocturnality in domestic cats for this area?
Line 112-113: Might be worth giving a reason that camera traps are useful study tool, otherwise this sentence seems like a filler. Alternately, this sentence doesn’t seem necessary.
Line 118: For the influence of domestic cat/cottontail rabbits on coyote occupancy I’d include the specific metric you were look at as influence (abundance/density/presence?)

Discussion
Similar to the intro, you discussion doesn’t really have a strong ‘so what’ aspect to it. I think there is a much more compelling angle which you touch that could be stressed a bit more. 1) results suggest coyotes are choosing to be in green spaces not selecting for habitat based on cat activity, 2) results somewhat suggest that it is the spatiotemporal overlap that makes cats prey to coyotes – thus keeping cats indoors, limiting outdoor time, or culling feral cats would reduce the number of cat depredations. Since the origins of this paper were management based, I think it’s only fair to tie in management implications.
Line 311: I think it’s a bit of a stretch to say this was a study determining the influences of coyote-cat conflict rather than just spatial/temporal patterning.

Line 316: I don’t think this should necessarily say the ‘proportion of green space within Culver City’ as that’s not really what you tested. Instead perhaps just rephrase as coyote occupancy was positively correlated with increased level of green space’ or something similar.

Figures
Figure 1: Should have city boundary on the map. Additionally might be nice for this to be a satellite image so that viewer can assess the spatial variability visually a bit better.

Experimental design

I have some fairly substantial concerns about the experimental design choices in picking the layout of the cameras which are elaborated in the uploaded PDF.

Materials & Methods
I have some fairly substantial qualms about the methodology used to place cameras which makes me question the validity of the results. I understand that you are looking at a very fine scale here, but the cameras simply seem to be too close together without a sufficient sample size to compensate for their proximity. I think a lot of your results may be biased by the ways in which the variables were defined and that original sampling schematic and choice to put cameras in areas where there were high densities of both sightings and near places where cats were being killed. If you are attempting to understand where coyote-cat conflict is going to happen, I’d imagine putting cameras only in the spots where we know coyote-cat conflict has happened likely isn’t going to reveal a bunch. Apologies if I’ve miss-understood the layout of the design, but even in looking at a map of the study design, the layout of the 20 cameras does not seem like it would capture a high enough variability to really be able to address many of you questions that you seek to ask on the spatial level. I imagine that the temporal component of this project is reflective of the study system, but that the spatial component may not be.
Line 135: You mention the cemetery as a major green space, but don’t actually have any cameras there, so I don’t think its to your benefit to mention or even stress how sizeable and potentially important this cemetery green space is. I know you placed cameras in places with higher sightings of coyotes, but I’d imagine that skewed your results to where there were more people present to observe them, rather than to where coyotes were present across the landscape which does make me have some question of the study design.
Line 147: Personal preference, but I’d change ‘transportation’ to ‘connectivity’
Line 180: I’m not sure I totally understand the justification for using coyote sightings reported to the police as a means for determining where cameras should go. I think this is potentially fairly biased especially given the people most likely to call into the police probably fall within very similar demographics that likely do not represent the range of the city.
Line 185: Might be nice to have a supplemental table/figure etc to show how frequent these reports were and also to know how they reported/evidence behind reports. From my experience, many people report missing cats, but don’t actually have evidence of their cat’s death let alone a cause of death. There are a lot of reasons that outdoor/feral cats disappear in urban areas that might not have anything to do with coyotes, though they are undoubtedly consuming cats as well.
Line 209: Unclear to me how many hexagons/what scale the tessellation is at within the 150m2 camera buffers
Line 215: I imagine some of the hexagons were reflective of some amount of both greenspace and residential space; if so, how was the classification decided? Also the binary of something being green space or not green space may not actually be that informative. We know that urban green spaces vary in quality/use based on the type of green space and what is present. I’m not sure this is the most informative variable and as the results suggest all you can really conclude is coyotes are using green spaces.
Line 230: Out of curiosity, why not attempt to individually identify domestic cats as has been done in other studies and a mark/recapture density estimate than choose an arbitrary 30min windows to calculate the number of cats? Unless they aren’t totally arbitrary in which case I think readers would appreciate a bit more detail.
Line 396: I’d also image that cats are present at almost all of your sites because the cameras appear to be very close together and are likely not only sampling the same coyotes (which you’ve justified), but also sampling the same domestic cats which could use justification.

Validity of the findings

No Comment

Annotated reviews are not available for download in order to protect the identity of reviewers who chose to remain anonymous.

·

Basic reporting

Instead of placing my review throughout multiple sections, I have included my entire review here.

# A review for:

**Spatiotemporal relationships of coyotes and free-ranging domestic cats as indicators of conflict in Culver City, California**


Note: This review is written in markdown format. I've also attached a pdf of the review if that is easier to read.

In this paper the authors examined the spatial variation in cat, cottontail, and coyote habitat use across Culver City, a city in southern California. I found the writing good, and the authors made a good case for the necessity of the research. Great job! From my reading of the manuscript, I've no big concerns with the study design or analysis. However, I would strongly caution the authors about the inference that is being made given the study design. I bring this up in length in the remaining review, but briefly, sites were selected such that coyote and cats were likely to occur there. This limits the inference that can be made, as these locations may not be representative of Culver City, on the whole. I do not think this invalidates the study at the least, but some care is necessary when considering the implications of the results in the discussion.

I'm waiving anonymity on this review, the authors are more than welcome to reach out to me if they need any clarifications about any questions or comments I have made below.

- Mason Fidino (mfidino@lpzoo.org)

## Introduction
* * *
### top-level thoughts

1. The writing here is great, and the flow of the introduction sets a clear path for the research. I've just got a few suggestions below that could improve certain parts that may require some additional explanation or clarification.

### line by line comments

Line 42-44: It is a little unclear what urbanization is having an affect on here. The first example pertains to the physical environment, while the remaining examples are likely associated to species in urban environments. Certainly, fragmentation can have positive and negative effects on wildlife, but it may help to make it more clear that all of these examples apply to species.

Line 49: This paragraph could be improved by adding a crystallizing point to the end of it (currenlty it ends on an example). Given the information you've provided a reader (through these great examples) what do you want them to take away? Does it make urban environments an exciting 'new' stage to answer questions about community dynamics? Or perhaps, as a result of these changing dynamics (which no doubt can vary within and among cities), much is still unknown about the extent to which species interact in urban environments? Given the end of paragraph 2 I'd suggest making the point about who much is still unknown about predator-prey interactions in urban areas. That way you can present the idea in paragraph 1 and nail down how coyote are an ideal species to better understand this question.

Line 50: Affected by what?

Line 71: are cats just in urban matrices? I'd think not, so maybe use something a little less specific. Likewise, you later lead into how we don't know about coyote cat interactions in urban green space (which isn't really the urban matrix) on line 91.

Line 91 - 92: Domestic cats and coyotes were part of our analysis of urban green species in Chicago (Gallo et al. 2017), so I'd change this sentence. Briefly, we found that cats were more likely to occupancy city parks whereas coyote where more likely to occur in other types of greenspace (e.g,. golf courses, cemeteries, and natural areas). This result actually seems quite different from your system given the large amount of spatial overlap among coyote and cat throughout Culver City, which is quite interesting!

```
Gallo, T., Fidino, M., Lehrer, E. W., & Magle, S. B. (2017). Mammal diversity and metacommunity dynamics in urban green spaces: implications for urban wildlife conservation. Ecological Applications, 27(8), 2330-2341.
```

Line 95: I wonder if you even need to introduce Culver City here? You could still be writing generally about coyotes & cats in southern California and bring up this Culver City information in the methods (which feels like the location in the paper that you can demonstrate how Culver city is an ideal place to study coyote & cat interactions). This would, of course, require some additional revision in the following paragraphs given that Culver City is brought up multiple times. Regardless of how the authors decide to tackle this, the sentence (as it is), is quite long and so some modification here would help.

Line 101-103: Coyote are often nocturnal in urban environments, I'd be specific about this (they are often crepuscular in more natural settings).

Line 103: Why do you predict cats will temporally overlap with coyote?

Line 106: Do cats really prey on cottontail rabbits? I think a lot of readers may wonder this. Or is this sentence trying to say that coyote may be selecting habitat based on other species (e.g,. rabbits)? I'm unsure.

Line 110-116: You are putting the cart before the horse by leading with this information in the last paragraph. It would be more clear to start with something like: "To better understand coyote and cat interactions in urban environments, we surveyed coyote and cats throughout Culver, City California using motion-triggered camera traps. Camera traps are... Using occupancy models (citation), we sought to determine..."


## Materials & Methods
* * *
### top-level thoughts

1. One assumption of standard occupancy models is that sampling locations are independent of one another. As such, if an individual is observed at site *A* it should not be available for sampling at site *B*. It is not the end of the world when this assumption is violated (which is often the case). It just means that the inference you make changes. Instead of making inference on occupancy, you are instead assessing relative patterns of habitat use. You are very close to making this point around lines 174-175, but it would perhaps help to define what you mean by occupancy for this study (perhaps in the statistical analysis section).

2. If camera trapping locations were informed by the presence of coyotes, then the estimated relationships should end up positively biased (because it is assessing occupancy in areas where coyote are reasonably expected to be). Likewise, camera trap sites were selected to be near where cat deaths were reported. Both of these no doubt modifies that type of inference that can be made, as the locations are not a random sample from Culver City, but rather at a subset of it where 1) coyote are expected to occur and 2) cat deaths have been observed. I think that the aim of the study provides reason for why this was necessary, but this limits the generality of the results (and as such should be acknowledged).

3. I'm confused about the 50m vs 150m buffer explanation. Is it that cameras were kept at a minimum of 50m apart? If so, that may be a more precise explanation for that.

4. What was the secondary sampling unit in the occupancy analysis? Cameras were active for 6 months, looking at `peerj-69611-InputFile_Coyotes.txt` there are 7 secondary sampling occasions. To add to the confusion figure 2 is apparently created with data across a 7 year period. I hope that this single-season occupancy model was not parameterized with data across a 7 year period (with each year as the secondary sampling unit).

5. The green space metric was z-transformed, could you provide the mean and sd of those data too? That would help with interpretation.

6. Eight models were fit to the data using all possible combinations. However, I'm uncertain which covariates were included on occupancy and which were included on detection. I can see this in the R scripts provided, but it may help to say something to the effect of "Given 2 possible occupancy covariates (x and y) and 1 possible detection covariate (z), there are 8 model combinations, including the null model." around lines 246.

### line by line comments

Line 149: *Camera Trap Analyses* feels like the wrong header for this section (given that there is a *Statistical Analyses* header as well). Perhaps *Sampling* or *Camera Trap Study Design*?

Line 168: What size buffer zone?

Line 169: Often this type of sub-sampling is not necessary, see Zuckerberg et al (2012, 2020). However, your buffers here are quite small, and so the locations are still almost on top of one another, so perhaps sub-sampling is a good idea?

```
Zuckerberg, B., Desrochers, A., Hochachka, W. M., Fink, D., Koenig, W. D., & Dickinson, J. L. (2012). Overlapping landscapes: A persistent, but misdirected concern when collecting and analyzing ecological data. The Journal of Wildlife Management, 76(5), 1072–1080. https://doi.org/10.1002/jwmg.326

Zuckerberg, B., Cohen, J. M., Nunes, L. A., Bernath-Plaisted, J., Clare, J. D., Gilbert, N. A., Kozidis, S. S., Maresh Nelson, S. B., Shipley, A. A., Thompson, K. L., & Desrochers, A. (2020). A review of overlapping landscapes: Pseudoreplication or a red herring in landscape ecology? Current Landscape Ecology Reports, 5, 140–148. https://doi.org/10.1007/s40823-020-00059-4
```

Line 203: See the Zuckerberg citations above about how overlapping buffers don't especially matter.

## Results
* * *
### top-level thoughts

1. When including your beta coefficients in the text of the results, you may as well also add the 95% CI's too (\beta = 2.18, 95% CI = lo#, hi#). This could completely remove tables 2 and 3, which are not especially needed. This should be done with regression coefficients and occupancy estimates throughout the entire results section.

2. There is no reference to the model intercepts here, which are actually quite helpful given the mean-centering of the covariates. Given the low number of covariates in general, it may make sense to do something like:

- say what the average occupancy of species x is (which is just the intercept value back-transformed to the probability scale via the inverse logit link).

- Following that, you can say that occupancy increased or decreased with increasing covariate y.

This lets you get all your information out to the reader. How common is the species. How does their distribution change. In fact, you do this for cottontail rabbit, which is great!




### line by line comments

Line 270: The highest ranked coyote occupancy models... There were multiple model fit to different species, so leaving coyote to the end of the sentence makes this awkward.

Line 278-280: Are you comparing variable importance between the occupancy and detection models? That's not really something that is done (they are separate processes). Likewise, if I recall, the standard turn of phrase is variable importance weight, not variable important value.


## Discussion
* * *
### top-level thoughts

1. The aims and study design do not line up as well with the discussion of the results. As I brought up in the methods section, camera traps were located in areas were coyote and cats are expected to occur. Yet, the discussion is more focused on the interactions among coyote and cats, on the whole, throughout Culver City. I think the points being made here are valid, but some caution is warranted here. Likewise, this may be something you could perhaps lean in to and capitalize on. It seems like the city has a good idea of where coyote are, and where cat deaths occur. How do the insights you learned here help us better understand these areas throughout the city?

Essentially, the study design made it so you were less likely to have areas where 1) coyote are but cats are not and 2) cats are but coyote are not. You tried to ensure both were going to be present, and so your inference should be conditional on this.

2. Perhaps one thing you should try to get ahead of, and maybe this is better placed in the methods, is that there are a multitude of occupancy models for multiple species (e.g., a dominant predator and subordinate prey species). Your sample size no doubt would not allow for fitting such models, and so you opted to use a metric of relative activity instead. One occupancy model that estimates co-occurrence that comes to mind is Waddle et al. (2010), but there are countless others. I leave it to the authors to decide whether or not they feel it is a good idea to try and get ahead of this, as other readers who are familiar with occupancy modeling may wonder this.

```
Waddle, J. H., Dorazio, R. M., Walls, S. C., Rice, K. G., Beauchamp, J., Schuman, M. J., & Mazzotti, F. J. (2010). A new parameterization for estimating co‐occurrence of interacting species. Ecological Applications, 20(5), 1467-1475.
```


### line by line comments

Line 316-318: Certainly, every discussion needs a caveat section, but don't put it at the end of the first paragraph! This is where your discussion of the main takeaways from your study should go. Leave the caveats for later in the discussion.

Line 348: There is a big difference between statistical significance and biological significance. Given the overlap figure the authors provided, it still appears that there is substantial temporal overlap among these species. Certainly, the peaks of activity may vary among species, but there is still about 0.70 overlap.

Line 372-382: I wonder how clustered in time the cat events were. Given the relatively long sampling window (6 months) it is possible that cat activity was not 'constant' at the site, and they simply had a lot of activity on just a smaller number of days.

Line 383: Again, this is conditional on your study design.

Line 411: Here too. The study design was focused on areas where the species were likely to overlap, as such it is difficult to make such a statement. Certainly, when they do overlap this sounds plausible, but the extent to which they overlap would require a different study design.


## Figures
* * *
Figure 2. If the data come from December 2019 to June 2020, why is this figure a heatmap over a 7 year period?

Experimental design

see above

Validity of the findings

see above

Reviewer 3 ·

Basic reporting

I think the basic reporting is clear and the article is really well written. The authors have done a great literature review, but rely heavily on the works of Kays and Gehrt throughout. I suggest they look into the works of Ashley Gramza and the following paper and references within: https://link.springer.com/article/10.1007/s11252-020-01026-x.

Objectives, research questions, and hypotheses were stated.

Experimental design

This research was well conducted. The authors acknowledge the very small spatial extent of the study area and the sample size (number of sites) and do a good job at justifying their study design in regard to these constraints. They also do a great job at using correct terminology and use the appropriate methods for a study of this nature.

I list a few concerns that I do have about the study design.

Line 178: This indicates that cameras are in locations where coyotes are known to occupy. This adds a level of bias in the analysis. However, on Lines 183-189, it seems cameras were also placed in random green spaces. Is it a mix of known locations and random locations, or just sites where coyotes had been detected in the past? Please clarify.

Lines 203-205: Following reference might help justify that overlap is not that big of a deal:

Zuckerberg, B., Cohen, J.M., Nunes, L.A. et al. A Review of Overlapping Landscapes: Pseudoreplication or a Red Herring in Landscape Ecology?. Curr Landscape Ecol Rep 5, 140–148 (2020). https://doi-org.mutex.gmu.edu/10.1007/s40823-020-00059-4

Lines 209-2017: This is not totally clear to me. Were multiple hexagons created inside the 150-m hexagon and the proportion of smaller hexagons were counted? Or were the number of cells within a large hexagon (assuming the underlying data were rasters) calculated? I think more details are needed to clarify how this variable was calculated. What were the “number of hexagons counted”?

Line 230: How does number of pictures per 30 minutes line up with coyote detection histories? Was the final variable average number of cat photos per site? In an occupancy model you have site covariates and observation covariates, but number of cat photos per 30 minutes doesn’t really line up with either. Please clarify.

Lines 234-244: If you are using AIC model selection you can have correlated variables in the model set, just not in the same models. NumRab could be included in the model set as long as it’s not in the same model as green space. I think its worthy of exploring as rabbits are an alternative prey source other than cats.

246-247: I suggest citing your model table here so the reader knows to go look at the different model combos.

Lines 271-274: I suggest not model averaging. The greenspace covariate comes out in the two top models and those two models are greater than 2 delta-AIC values from the next additive model. Just interpret the beta coefficient of your top model. There is debate around model averaging (see Cade 2015, Ecology and Banner and Higgs, 2016), and, to me, this analysis could avoid that debate because the top models are pretty straightforward. Same goes with the rabbit model. The results are very clear, no need to muddle the results with model averaging.

Validity of the findings

Again, recognizing that small spatial extent of the study and the small sample size of sampling sites, I do believe the results are sound.

My major comments about the results are as follows:

Results in general: It is important to keep the occupancy model results and the detection model results separate, especially for those readers that are not familiar with occupancy models. The VIF (Line 280) and the results of the rabbit model discuss CamHeight in the same context as the other covariates (Line 295) put on occupancy. This could confuse readers as CamHeight was never a covariate for occupancy. It will help the reader follow if the parameters are explicitly described in the context of the response variable.

Line 302: Is 0.90 a common cut off for “low temporal overlap”?

In regards to the Discussion and Conclusions I do have major concerns about the inference that is made. The most important concern is that the discussion topic of optimal foraging theory leans too heavily on this thin connection between coyote occupancy and cat “abundance”, when cat abundance was never really measured. In regards to the variable, what if there are lot of cats, but they are less active when coyotes are around? This would return fewer pictures of cats. What if there are only a few cats, but their activity is concentrated around the camera site? This would return many pictures, but true abundance would be low. In regards to the results, what if CatNum had no effect, because Culver City coyotes don’t really prey on cats? Or what if CatNum had no effect because coyotes prey on them heavily, but humans are constantly replacing their “missing” cat? Then the results would seem to show that there are always cats around, but really there is high turnover of individuals?

I personally think the inference here about the predation of cats by coyotes is too far of a reach given the data. Especially for it to be the center of the narrative. Cats didn’t show up as a variable in the top models, so at best, the results say that their distributions overlap by chance. Really, the results show the activity of cats does not influence the presence of coyotes.

The results do, however, indicate that available habitat matters more than prey availability. Perhaps urban coyotes are good at finding food regardless of the availability of prey, so what they are really limited by is greenspace. Gallo et al 2017, Journal of Applied Ecology, sort of suggests this by showing rabbits were not spatially avoiding coyotes nor displaying more vigilance when coyotes were around. Adding rabbit photo numbers in the model selection procedure would also help answer this angle. If rabbits also have an insignificant effect, compared to greenspace, then perhaps the above hypothesis is correct.

Other concerns are as follows:

Line 315-318: Technically, coyote occupancy was best predicted by greenspace compared to the availability of cats or no covariates at all (null). With model selection, the top model explains more information relative to the other models. Once could totally miss the real story if they don’t include the right variables. The analysis didn’t include any measurement of human activity (e.g., population density) or the built environment (except assuming the inverse of greenspace was the built environment). Urban coyotes are susceptible to roads and people. I would suggest, at minimum, adding some metric of people and roads into your model set to make stronger inference about greenspaces.

Line 3265: I suggest not using the term “selecting”. Habitat use was measured, but not habitat selection.

Line 340: I think there is an opportunity to include rabbits in the coyote analysis, I also think coyotes should be included in the rabbit analysis in the same way.

Line 348: The cut off of 90% overlap being significantly different needs to be cited.

Line 374: I suggest calling this cat activity and not relative abundance. The covariate is more of a measure of activity as one cat could be in many of the observations. Therefore, there could be low cat abundance, but at the species level there is lots of cat activity since at least one cat is at the site a lot.

---

## Round 0.2 · Minor Revisions

Thank you for your thorough and thoughtful revisions. Both reviewers are pleased by your work and offer a small number of relatively minor revisions for you to address. I look forward to receiving a revised version of your manuscript.

Reviewer 1 ·

Basic reporting

no comment; the manuscript is much improved from the first edition.

Experimental design

Authors have done a good job of explaining methodology and limitations to the study & interpretation of the results

Validity of the findings

no comment

Additional comments

General comments:
The authors have significantly improved the manuscript. The interpretation of results appears reasonable.
You refer to ‘exceptionally’ or ‘abnormally’ high rates of conflict, but don’t provide evidence that the levels of conflict are actually higher than in other regions; additionally, because I think there is an important distinction between perceived and actual conflict. Some people perceive a coyote looking at a cat as a conflict, but no conflict actually occurs. From personal experience in going through coyote-conflict reports, there’s a pretty high amount of these false conflicts.
Introduction
Line 77: as  is
Line 106: Can you elaborate on what ‘particularly high rates of domestic cat depredation’ means or how this was determined/compares to other places? Also how are the depredations determined? Coyotes could simply be eating totally stray cats or cats that were hit by cars, or cats can die/disappear for many many other reasons.
Line 137: Can you please elaborate on how cottontails may mediate coyote-cat conflict
Materials & Methods
Line 156-157: Don’t really think you need to add that this oil field is one of the most productive
Line 193: May want to change ‘citizen’ to ‘community’; debate goes both ways, but worth considering.
Results
I think it might be nice to give an idea of how many cat/coyote/rabbit images were collected
Line 310: p is detection correct? Would be nice to refer to it a detection probability rather than p for people who are less familiar with occupancy modeling similar to what you do with psi
Discussion
Line 450-451: I don’t actually think there is super strong evidence to suggest this. The way the Larson studies are done leave a lot of potential room for error in my personal opinion.
Figures
Figure 1: Can you change the outline of the city to a color that stand out on the map a bit more? Similarly, it is sort of difficult to visually distinguish the camera site points
Figure 2: would be good to have a map legend

·

Basic reporting

# Review for:

*Spatiotemporal relationships of coyotes and free-ranging domestic cats as indicators of conflict in Culver City, California*


I've placed my entire review in this box. Great job on the revisions. I just noticed a few minor things that could use a little tweaking to increase clarity. Also, table 3 looks to be incorrect (it seems like it should be about the activity analysis but instead it has the greenspace slope terms from the occupancy model).

- Mason Fidino

## Abstract
* * *
### Line by line comments

Line 22: I think the "Los Angeles County, California" part is a little too specific. For example, is it known why cats comprise a large majority of coyote diets in all other southern california counties but Los Angeles county? If not, I'd probably just change the end of that sentence to something like "...cat depredation by coyotes in this region."

Line 28: "...across six months in {insert year here}..."


## Introduction
* * *
### Top-level thoughts

1. I've got no comments here, I often found myself nodding along in agreement with the logical flow of the information presented. Well done!


## Methods
* * *
### Line by line comments

Line 178: Could you add an "(e.g., {insert examples ehre})" following "limitations in the urban landscape." That turn of phrase is a little vague.

Line 181 - 192: Wonderful explanation here.

Line 246: Change the end of the sentence to "... and standardized these covariates to have a mean zero of and unit variance."

Line 289: Cite R here as well. 4.0.3 came out in 2020 so the citation here would be "R Core Team, 2020"

## Results
* * *
### Top-level thoughts

1. I'd check with the journal guidelines about the number of significant digits to include. Right now it's three, sometimes two are suggested.

### Line by line comments

Line 321-324: Include the 95% CI as well instead of just saying it bounds zero.

## Discussion
* * *
### Top-level thoughts

1. Great revision of the discussion.



## Tables & figures
* * *
Table 3: I'm confused why table 3 has a \Psi on the top when it should be about the activity analysis. Line 342 indicates this while Table 3 looks like it's the slope terms from the occuapancy model.

Experimental design

see above

Validity of the findings

see above

---

## Round 0.3 · accepted · Accept

Thank you for your thoughtful and thorough responses to the reviewers. I look forward to your publication.